# Openness to Mental Health Information and Barriers to Accessing Care Among Midwestern Farmers

**DOI:** 10.3390/ijerph23010027

**Published:** 2025-12-24

**Authors:** Courtney Cuthbertson, Samantha Iwinski, Asa Billington, Josie Rudolphi

**Affiliations:** 1Department of Human Development and Family Studies, University of Illinois at Urbana-Champaign, Urbana, IL 61801, USA; cuthbert@illinois.edu (C.C.); iwinski2@illinois.edu (S.I.); 2Department of Counseling, Human Development and Family Science, University of Vermont, Burlington, VT 05405, USA; asa.billington@uvm.edu; 3Department of Agricultural and Biological Engineering, University of Illinois at Urbana-Champaign, Urbana, IL 61801, USA

**Keywords:** mental health, barriers to care, farmers, support

## Abstract

**Highlights:**

**Public health relevance—How does this work relate to a public health issue?**
Midwestern farmers face significant barriers to mental health care, which vary by demographic and attitudinal factors.

**Public health significance—Why is this work of significance to public health?**
Understanding which populations are least open to mental health information helps prioritize outreach to those at greatest risk of unmet needs.

**Public health implications—What are the key implications or messages for practitioners, policy makers and/or researchers in public health?**
Public health education and health promotion efforts should be tailored to encourage openness to information and help-seeking.

**Abstract:**

Agricultural producers experience elevated stress, limited mental health access, and cultural norms that can discourage help-seeking. This study examined farmers’ preferences for receiving mental health information and the barriers that impede care. Data came from a regional needs assessment of 1024 producers across 12 Midwestern states who completed online or paper surveys, including questions on willingness to seek or receive information and the 30-item Barriers to Access to Care Evaluation. Analyses included descriptive, bivariate, and multivariate methods to explore demographic and behavioral predictors. Results indicated that while 74.1% were open to receiving mental health information, notable proportions were unwilling to seek (27.8%) or receive (28.4%) it, and 18.7% were unwilling to do either. Preferred sources were medical providers, mental health professionals, and family members, with agricultural retailers least favored. Women, younger producers, veterans, those with mental health symptoms, and individuals with higher education, anxiety, or depression showed distinct patterns of openness and barrier endorsement. Attitudinal barriers were the most common across groups. Findings highlight the importance of culturally relevant approaches that leverage trusted messengers, reduce stigma, and tailor interventions to specific subgroups to strengthen mental health outreach in agricultural communities.

## 1. Introduction

Agricultural producers experience significantly poorer mental health outcomes compared to the general population, with elevated rates of both anxiety and depression. Prevalence estimates for anxiety and depression among this population range from 18.7% to 71% and 13.2% to 53%, respectively, with one study reporting that approximately one in eight producers (13.2%) met the clinical threshold for depression and nearly one in five (18.7%) met criteria for generalized anxiety disorder [1,2,3,4,5].

Despite the documented need, agricultural producers face barriers to mental health support. Cultural norms that emphasize independence, toughness, and self-reliance remain deeply rooted in farming communities and discourage help-seeking [6,7,8]. Public and self-stigma around mental illness, especially among older adults and men, continues to act as an obstacle [9]. Barriers to care include fear of peer judgment, concerns about social image, and limited trust in available services [10,11]. Structural and logistical challenges compound these issues, including shortages of rural providers, lack of anonymity, limited mental health literacy, long travel distances, and time constraints driven by farming’s demanding schedule [12,13,14]. Researchers have also found that this stigma can be further compounded by a misalignment with federally qualified health centers’ locations and farming regions and limited culturally responsive programs for some communities in Hawai‘i [15]. Farmers frequently express that the farm must come first, and the perception that mental health professionals do not understand agricultural life only deepens the divide between need and access [11,16]. Recent literature has found that these barriers may come from farmers putting their farming operations before their own physical and mental health. In Belgium, Flemish farmers were rooted in a cultural pattern called “passive endurance,” a mix of hard work, toughness, self-reliance, resignation, powerlessness, and a belief that challenges must be endured. This outlook discouraged help-seeking, as many preferred to “just keep working” and doubted the usefulness of aid, especially mental health services [17]. Although digital tools offer new pathways for support, challenges such as low internet access, limited digital literacy, and discomfort with virtual platforms persist [14,18]. Despite increasing attention to mental health in agriculture, a complex web of personal, social, and structural barriers continues to impede access to care.

Many outreach efforts seek to improve mental health outcomes in agricultural communities by enhancing mental health literacy, promoting awareness of symptoms, communication strategies, and referral resources. These include the availability of telephone helplines and hotlines [19], training programs aimed at increasing mental health literacy among agricultural community members [20,21], and programs to reduce stress and improve coping skills among agricultural producers. Researchers often suggest improving mental healthcare delivery to farming populations by addressing accessibility, stoicism, stigma, dual roles, and community trauma [22].

However, less is known about farmers’ openness to receiving mental health information from various sources, or how they perceive barriers to accessing care. In particular, it is unclear who farmers prefer to receive such information from, how they want it delivered, and what barriers might prevent it from being effectively shared. This study addresses these gaps by examining both the willingness of Midwestern farmers to engage with mental health information across different sources and the barriers they experience in seeking care.

## 2. Materials and Methods

This study utilized data from a regional needs assessment of agricultural producers across the Midwestern region of the United States, conducted by the North Central Farm and Ranch Stress Assistance Center. The goal of the regional needs assessment was to examine the prevalence and patterns of physical health, mental health, and substance use, of Midwestern agricultural producers. This study received approval from the Institutional Review Board of the University of Illinois Urbana-Champaign (protocol number 21640, 2 April 2021).

### 2.1. Participants

Potential respondents were eligible to participate in the study if they were at least 18 years of age, resided in one of 12 Midwestern states, and were actively farming and/or ranching at the time of data collection.

### 2.2. Recruitment & Data Collection

US Farm Data (formerly Farm Market iD), which is an organization that houses information regarding the addresses of farms and farmer demographics collected from USDA Census of Agriculture surveys, as well as from publicly and privately sourced data [23], provided the addresses of 15,000 randomly selected agricultural producers in the Midwestern region. Paper surveys were then mailed to these addresses in three steps, using a modified Dillman approach to encourage survey response [24]. The first step included mailing an introductory letter that included the study’s objectives, commonly asked questions about the research, and contact information for the study team and the Institutional Review Board. The initial mailing, sent in the summer of 2021, included the survey, a list of mental health resources (talk and text hotlines, crisis line numbers, and websites aiding individuals in crisis or distress), and a postage-paid return envelope. Approximately six weeks after first contacting agricultural producers, postcard reminders about the survey were mailed. Six weeks after the reminder postcards were mailed, a final mailing included a revised introductory letter, another copy of the survey, and a postage-paid return envelope was sent to agricultural producers. Of the 15,000 agricultural producers contacted, 254 mailings were returned as undeliverable, 180 individuals were determined to be ineligible, and 36 actively refused participation. The remaining non-responses were from individuals who did not reply. A total of 1110 eligible and complete responses were included in the final sample. No participants requested to withdraw after submission. A team of trained undergraduate and graduate students entered responses from the paper surveys into a data management system to facilitate data cleaning and analysis.

### 2.3. Measures

#### 2.3.1. Openness to Mental Health Information

The survey included one question about how open participants were to receiving mental health information overall, with response options of “very open,” “somewhat open,” and “not at all open.” Participants were asked whether they were interested in receiving mental health information from 19 specific sources, with “yes” and “no” response options, as well as whether they were likely to seek out mental health help or assistance from each of the 19 sources. Those sources included agricultural bankers, retailers, commodity groups, licensed medical providers, licensed mental health providers, mental health organizations, friends, family, and spiritual leaders.

#### 2.3.2. Barriers to Care

The Barriers to Access to Care Evaluation (BACE; [25]) asked participants to rate how much each of the 30 items had ever stopped, delayed, or discouraged them from seeking mental health care. Response options were on a four-point scale from “not at all” to “a lot.” The items create one overall BACE score (α = 0.95) as well as three subscales for attitudinal (α = 0.87), instrumental (α = 0.82), and stigma-related (α = 0.93) barriers. Attitudinal barriers included statements like “thinking that professional care probably would not help.” Instrumental barriers included statements such as “having no one who could help me get professional care.” Stigma-related barriers included the statement “concern that I might be seen as ‘crazy,’” among others.

#### 2.3.3. Mental Health

The survey included the PHQ-4 to measure anxiety and depression symptoms. Participants responded on a four-point scale from “not at all” to “nearly every day” considering their experience over the past two weeks, including “feeling nervous, anxious, or on edge,” and “feeling down, depressed, or hopeless.” Participants were also asked whether they had ever been diagnosed with anxiety or depression.

#### 2.3.4. Physical Health

Participants were asked whether they had ever been diagnosed with each of 12 physical health conditions, including asthma, arthritis, cancer, chronic pain, COPD, dementia and/or Alzheimer’s, diabetes, hearing loss, heart condition, high blood pressure, high cholesterol, and obesity.

### 2.4. Data Analysis

We conducted descriptive statistics to summarize key variables and used chi-square tests to examine associations between categorical variables. Given that the dependent variables were not normally distributed, we employed non-parametric methods to assess group differences. Specifically, Kruskal–Wallis H tests were used to determine whether statistically significant differences existed between groups. Analyses were conducted using SPSS Version 29.

## 3. Results

The sample was predominantly male (92.1%), white (98.8%), and heterosexual (99.2%). Most respondents were married (83.9%), and about one-third (35.6%) had earned a bachelor’s degree or higher. Approximately 17.1% were military veterans, and a majority (82.4%) identified as moderately to very religious. Most participants (87.7%) were involved in conventional farming, 2.2% reported practicing organic farming, and 10.1% engaged in both. Nearly three-quarters (74.6%) identified themselves as their farming operation’s principal or primary owner/operator. Field crops were the most common primary commodity (69.4%), followed by beef (16.2%), dairy (3.0%), and other commodities (11.4%). Participants reported a range of chronic health conditions. The most common diagnoses included high blood pressure (44.7%), high cholesterol (36.2%), hearing loss (28.8%), and arthritis (26.0%). Other frequently reported conditions were obesity (10.1%), chronic pain (16.3%), and diabetes (13.4%). Mental health conditions were also reported, with 8.0% of participants indicating a diagnosis of anxiety and 8.2% reporting depression. Fewer participants reported asthma (6.9%), cancer (15.4%), heart conditions (15.1%), or COPD (3.6%).

Regarding openness to receiving mental health information, 19.6% of participants indicated in general they were very open, compared to 55.3% who said somewhat open, and 25.1% who said they were not at all open. Results indicated that while 74.1% were open to receiving mental health information, notable proportions were unwilling to seek (27.8%) or receive (28.4%) it, and 18.7% were unwilling to do either. Participants reported the least willingness to receive and to seek information from agricultural retailers (6.3% and 3.2%, respectively), and the most willingness from licensed medical professionals (59.6% and 58.2%, respectively). Spouse or family was the second highest, with 56.6% willing to receive and 55.2% willing to seek mental health information from these sources. A notable portion of the sample reported unwillingness to engage with mental health information: 28.4% were not willing to receive it from any source, and 27.8% were not willing to seek it out. Additionally, 18.7% expressed no willingness to either receive or seek out mental health information. Figure 1 displays a chart of willingness to receive and seek information from each of the 19 sources.

Openness to receiving mental health information was associated with gender as a significantly higher proportion of men were not open at all to receiving mental health information (25.7%) compared to women (14.9%; χ^2^ = 3.826, *p* = 0.05). A greater proportion of participants with a bachelor’s degree or more education (82.6%) was somewhat or very open compared to 70.8% of participants with up to an associate’s degree (χ^2^ = 14.960, *p* < 0.001). Religiosity was significantly associated with openness to receiving mental health information, with 80.5% of participants who were very religious being somewhat or very open compared to 72.9% of participants who were not religious (χ^2^ = 6.195, *p* = 0.013). Having an anxiety diagnosis was significant (χ^2^ = 9.30, *p* = 0.002) as 90.3% of those diagnosed were somewhat or very open to receiving mental health information, a higher proportion compared to those without an anxiety diagnosis (74.1%). Having a depression diagnosis was similar; 94.2% of those with a depression diagnosis were somewhat or very open to receiving mental health information compared to 73.8% of those without a depression diagnosis (χ^2^ = 14.226, *p* < 0.001). Age, being married, being a veteran, having a disability, farm role, anxiety symptoms, and depression symptoms were not significant.

Mean total BACE scores are presented in Table 1. The overall barriers to care mean score was 0.257 (SD 0.42). Barriers to care include subscales for stigma-related (Table 2), instrumental (Table 3), and attitudinal barriers (Table 4). The mean for attitudinal barriers (0.338, SD 0.49) was highest, followed by stigma-related barriers (M 0.248, SD 0.48), and instrumental barriers (M 0.146, SD 0.33). Perceived barriers to mental health care varied significantly across several participant characteristics. Women in the sample reported significantly greater instrumental and attitudinal barriers to care compared to men. Higher levels of alcohol use were associated with greater perceived barriers to care, particularly attitudinal barriers. Participants whose symptoms met criteria for anxiety disorder or major depressive disorder, as well as those diagnosed with either condition, reported significantly higher perceived barriers across all subscales. In contrast, individuals who were not open at all to receiving mental health information perceived significantly fewer barriers than those who were somewhat or very open. No significant differences were found in perceived barriers to care based on farm role or marital status.

## 4. Discussion

The goal of this study was to examine Midwestern farmers’ openness to mental health information and their perceived barriers to accessing mental health care. This study found that participants were most willing to both receive and seek out mental health information from medical providers, spouses/family members, mental health providers, religious leaders, and friends. It is notable that participants’ preferred sources of mental health information were community members and people close to them. Additionally, across all sources of mental health information, participants were more willing to receive information than to seek it out. Together, these findings suggest that an approach to farmers’ mental health care that sends information through farmers’ social networks would be especially beneficial.

Examining demographic patterns of openness to mental health information demonstrated that participants with diagnosed anxiety or depression were more open to receiving and seeking out mental health information. This finding may be attributed to the likelihood that receiving a diagnosis for a mental health condition inherently requires an individual to interact with mental health providers and information. Furthermore, individuals with diagnosed mental health conditions are likely more motivated to access information about mental health to address their known mental health needs. Participants who endorsed higher levels of religiosity were also more open to receiving and seeking out mental health information. This finding may be attributed to perceptions of topics such as mental and emotional well-being as more appropriate to address in religious settings, which could create more comfort in receiving or seeking out mental health information. Studies have found that religion may offer positive long-term mental health effects through participation in public religious activities and the importance of religion to oneself [26]. Although more research is needed to determine the nature of the relationship between religiosity and openness to mental health information among farmers, this finding suggests a promising avenue for providing mental health support for farmers in religious settings. This includes USDA faith-based initiatives and opportunities within the Regional Farm and Ranch Stress Assistance Network. In addition, men and participants with less education were less open to receiving or seeking out mental health information. A review by Seidler et al. (2016) found that greater conformity to traditional masculine norms is associated with reduced help-seeking behavior in men, which may influence how individuals recognize and express symptoms, their attitudes toward seeking help, and how they manage mental health challenges [27].

A notable proportion of the sample were not willing to either receive or seek out mental health information from any source. Additionally, participants who were not open to receiving or seeking out mental health information reported fewer barriers to accessing mental healthcare. These findings elicit the question of how to provide mental health support for farmers unwilling to receive the information. Research suggests creating community-based programming to accommodate farmers’ specific occupational and cultural needs [28]. More qualitative studies on farmers’ perceptions of stress and mental health are needed to understand their needs and how to address them, and deliver mental health services and education to meet farmers where they are.

Some groups of participants reported greater perceived barriers to accessing mental healthcare. Barriers to accessing mental healthcare were organized into three categories according to the BACE measurement, which included stigma-related, attitudinal, and instrumental barriers. Stigma-related barriers refer to concerns someone may have about how others may negatively perceive their accessing of mental healthcare, attitudinal barriers refer to discomfort with discussing mental health with others, and instrumental barriers refer to tangible obstacles in obtaining mental healthcare. Women indicated experiencing more instrumental barriers and attitudinal barriers in accessing mental healthcare than men. The concept of emotional labor in local food systems offers a valuable lens for understanding this disparity in agricultural settings. As Som Castellano (2016) explains, women are often expected to care for their families and the emotional well-being of others in their community, all while contributing to or managing farm operations [29]. These multiple demands create a layered emotional strain that can make it especially difficult for women to prioritize their mental health, which can be further compounded by limited resources, geographic isolation, and competing responsibilities in farming contexts. Such conditions may reinforce instrumental barriers (e.g., lack of time, transportation, or service availability) and attitudinal barriers (e.g., guilt over self-care, discomfort with mental health conversations). Women’s experiences in agriculture are often shaped by intersecting responsibilities and identities, suggesting the need for further research that captures the layered nature of their roles [30,31,32].

Additionally, participants who reported higher recent alcohol use reported higher attitudinal barriers. This finding may be attributed to the use of alcohol as a coping strategy. If participants avoid seeking mental healthcare due to discomfort with discussing their emotions, they may turn to alcohol as an alternative coping mechanism [33,34,35,36]. Finally, participants who experienced anxiety or depression symptoms that met the diagnosis threshold or had diagnosed anxiety or depression reported greater instrumental, attitudinal, and stigma-related barriers to accessing mental healthcare. Participants experiencing more frequent or severe symptoms of depression or anxiety may be more likely to seek mental health care, yet they could also encounter greater barriers to accessing it. Individuals experiencing depression and/or anxiety may face barriers to accessing mental health care due to limited social functioning, lack of a usual source of care, or insurance coverage, which are factors shown to impact utilization of mental health services [37].

Agricultural producers face many barriers to accessing care, particularly for mental health issues. The findings indicate a significant variance in openness to receiving mental health information, highlighting a hesitancy that may stem from the cultural norms of stoicism and self-reliance. Such values, while fostering a sense of resilience and independence, may simultaneously discourage help-seeking behaviors and the acknowledgment of mental health struggles [8]. Barriers to mental health support include strong norms of self-reliance and a tendency to downplay or distance oneself from mental health concerns [38]. Hopkins et al. (2023) identified cultural norms, normative beliefs about healthcare, and stigma surrounding mental health as key barriers inhibiting farmers from seeking the help they need [6]. This may reflect what Hopkins et al. (2024) describe as “farm identity,” a deep-rooted connection to farm work that often leads farmers to prioritize their operations over their own physical and mental well-being [30]. Stigma surrounding mental health is especially pronounced in rural communities, where close social networks heighten privacy concerns. Such stigma operates on multiple levels, including internally, by discouraging individuals from recognizing their own need for support, and externally, through community attitudes toward those who pursue mental health services. Among farmers, a preference for face-to-face interactions combined with distrust of health professionals further compounds these barriers to help-seeking [10,11,14,18]. In addition to stigma and provider distrust, some agricultural producers may perceive mental health care differently due to the stressors they face, such as weather variability, high input costs, or fluctuations in commodity markets. This may interact with perceptions that providers lack the lived experience to understand the agricultural context [14,18]. As a result, farmers often rely on alternative coping strategies shaped by both individual characteristics and broader community norms, particularly where stigma makes accessing formal treatment less acceptable [33,34,35,36].

### Limitations

This study has several limitations. First, findings are based on responses from agricultural producers in a 12-state region and may not apply to all people who work in agriculture. The sample also lacks diversity in terms of race/ethnicity, gender, and sexual orientation. The response rate was also lower than rates typically reported in mailed surveys targeting agricultural producers [2,39,40,41]. Nonetheless, many recent cross-sectional studies on stress, mental health, and substance use in farming populations have employed convenience, online, or snowball sampling techniques [42,43], which do not permit calculation of response rates or support generalizability. Moreover, lower response rates are a common challenge in agricultural research settings [44,45].

In addition to these concerns, there may be some sampling bias, as individuals who experience shame, distrust of healthcare systems, or mental health stigma may have chosen not to participate. Others who did respond may have done so with limited openness. Furthermore, the survey utilized Likert-scale items, which limited the depth of insight into participants’ experiences and perceptions and did not allow for open-ended responses that could facilitate qualitative analysis. Another limitation is the imbalance in distribution, with a larger number of male participants compared to female participants. Lastly, seasonal timing may have influenced the data, as symptoms of stress, anxiety, and depression—as well as perceptions of mental health—can fluctuate throughout the year based on environmental conditions, such as weather, planting and harvest cycles, and other seasonal stressors commonly experienced in agricultural communities.

## 5. Conclusions

Understanding how producers engage with mental health information is essential. This study highlights the variability in receptiveness among agricultural producers and the importance of trusted sources. Findings suggest that dissemination strategies should be tailored to reflect producers’ preferences and existing trust networks. By aligning outreach efforts with these insights, programs can more effectively connect producers to the needed resources, reducing barriers to support and fostering a more supportive agricultural community.

## Figures and Tables

**Figure 1 ijerph-23-00027-f001:**
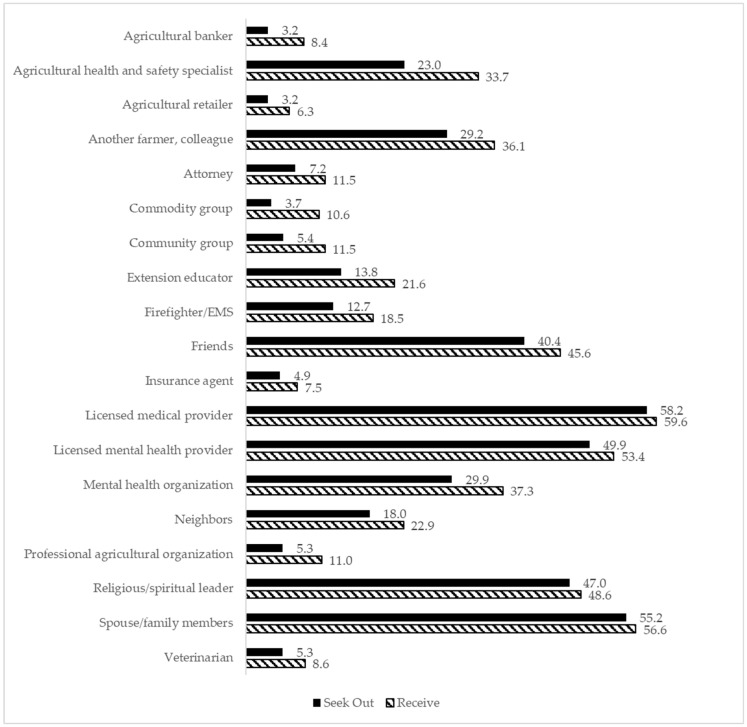
Percent of participants willing to seek out and receive mental health information from each of 19 sources.

**Table 1 ijerph-23-00027-t001:** Associations with Barriers to Care average score.

		*n*	Mean Rank	H	*p*	η^2^
Gender	man	948	510.09	3.723	0.054	0.00
	woman	81	572.49			
Age	64 and under	372	427.60	26.450	<0.001	0.03
	65 and older	401	349.34			
Education	Up to an associate degree	640	489.02	4.625	0.032	0.00
	Bachelor’s degree or higher	365	527.51			
Veteran status	No	861	530.33	12.656	<0.001	0.01
	Yes	171	446.86			
Alcohol use	Low risk	565	361.23	4.990	0.025	0.00
	At risk, high risk, possible AUD	175	400.41			
Anxiety symptom threshold	No anxiety	925	479.92	108.513	<0.001	0.10
	Probable anxiety	90	796.56			
Anxiety diagnosis	No	871	462.35	49.330	<0.001	0.05
	Yes	91	644.75			
Depression symptom threshold	No depression	944	486.54	54.251	<0.001	0.05
	Probable depression	60	753.63			
Depression diagnosis	No	877	459.15	65.681	<0.001	0.07
	Yes	82	702.95			
Open to MH Info	Not at all	204	329.31	40.388	<0.001	0.05
	Somewhat/Very open	629	445.44			
Open to MH Info	Not at all	204	162.36	24.059	<0.001	0.06
	Very open	164	212.05			

Note. Alcohol Use Disorder (AUD); Mental Health (MH); Information (Info).

**Table 2 ijerph-23-00027-t002:** Associations with Barriers to Care—Stigma subscale.

		*n*	Mean Rank	H	*p*	η^2^
Gender	man	936	506.14	1.045	0.307	0.00
	woman	80	536.07			
Age	64 and under	369	417.66	23.197	<0.001	0.03
	65 and older	396	350.70			
Education	Up to an associate degree	631	488.21	2.198	0.138	0.00
	Bachelor’s degree or higher	362	512.32			
Veteran status	No	850	552.43	12.420	<0.001	0.01
	Yes	169	447.51			
Alcohol use	Low risk	559	359.66	3.707	0.054	0.00
	At risk, high risk, possible AUD	174	390.57			
Anxiety symptom threshold	No anxiety	914	479.84	82.434	<0.001	0.08
	Probable anxiety	89	729.62			
Anxiety diagnosis	No	859	459.78	39.497	<0.001	0.04
	Yes	91	623.91			
Depression symptom threshold	No depression	937	485.17	46.456	<0.001	0.04
	Probable depression	59	710.16			
Depression diagnosis	No	866	454.93	66.252	<0.001	0.07
	Yes	82	677.91			
Open to MH Info	Not at all	203	331.01	42.277	<0.001	0.05
	Somewhat/Very open	622	439.76			
Open to MH Info	Not at all	203	163.55	25.607	<0.001	0.07
	Very open	163	208.35			

Note. Alcohol Use Disorder (AUD); Mental Health (MH); Information (Info).

**Table 3 ijerph-23-00027-t003:** Associations with Barriers to Care—Instrumental subscale.

		*n*	Mean Rank	H	*p*	η^2^
Gender	man	943	506.00	8.961	0.003	0.01
	woman	81	588.14			
Age	64 and under	371	420.44	28.884	<0.001	0.03
	65 and older	397	350.91			
Education	Up to an associate’s degree	636	493.56	1.784	0.182	0.00
	Bachelor’s degree or higher	365	513.96			
Veteran status	No	857	524.69	10.424	0.001	0.01
	Yes	170	460.11			
Alcohol use	Low risk	563	364.35	1.735	0.188	0.00
	At risk, high risk, possible AUD	174	384.03			
Anxiety symptom threshold	No anxiety	921	480.85	115.727	<0.001	0.11
	Probable anxiety	89	760.62			
Anxiety diagnosis	No	866	462.97	46.854	<0.001	0.05
	Yes	91	631.53			
Depression symptom threshold	No depression	940	484.41	75.762	<0.001	0.07
	Probable depression	60	752.61			
Depression diagnosis	No	872	461.09	55.755	<0.001	0.06
	Yes	82	652.04			
Open to MH Info	Not at all	204	351.55	28.423	<0.001	0.03
	Somewhat/Very open	624	435.08			
Open to MH Info	Not at all	204	163.97	28.596	<0.001	0.08
	Very open	164	210.03			

Note. Alcohol Use Disorder (AUD); Mental Health (MH); Information (Info).

**Table 4 ijerph-23-00027-t004:** Associations with Barriers to Care—Attitudinal subscale.

		*n*	Mean Rank	H	*p*	η^2^
Gender	man	940	505.43	4.102	0.043	0.00
	woman	80	570.02			
Age	64 and under	370	419.85	20.732	<0.001	0.02
	65 and older	398	351.64			
Education	Up to associates degree	634	484.62	5.363	0.021	0.00
	Bachelor’s degree or higher	364	525.41			
Veteran status	No	854	524.45	10.623	0.001	0.01
	Yes	169	449.08			
Alcohol use	Low risk	558	357.47	5.396	0.02	0.00
	At risk, high risk, possible AUD	175	397.40			
Anxiety symptom threshold	No anxiety	917	475.05	115.359	<0.001	0.11
	Probable anxiety	89	796.61			
Anxiety diagnosis	No	863	460.86	38.064	<0.001	0.04
	Yes	91	635.35			
Depression symptom threshold	No depression	938	482.99	56.905	<0.001	0.05
	Probable depression	59	753.50			
Depression diagnosis	No	870	457.23	55.088	<0.001	0.05
	Yes	81	677.62			
Open to MH Info	Not at all	204	337.50	32.705	<0.001	0.04
	Somewhat/Very open	625	440.30			
Open to MH Info	Not at all	204	164.25	19.861	<0.001	0.05
	Very open	163	208.71			

Note. Alcohol Use Disorder (AUD); Mental Health (MH); Information (Info).

## Data Availability

Data is not available to the public due to confidentiality.

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
