# Peer review of "Openness to Mental Health Information and Barriers to Accessing Care Among Midwestern Farmers"

_ijerph, 2025, doi:10.3390/ijerph23010027_

Round 1
Reviewer 1 Report
Comments and Suggestions for Authors
The manuscript describes a study aimed at understanding how agricultural operators report barriers to seeking mental health care and whether they are seeking information or are willing to receive information. The authors correctly assert that this is a difficult population to survey with typically low response rates. It is also a subject that many in the agricultural community do not wish to discuss. There are some questions about the methods that should be addressed.
The 30-item Barriers to Access to Care Evaluation Scale was used in this study, but wasn’t this study designed for those who are assumed to have a mental health condition for which treatment seeking is warranted?
Why was the BACE used and not the Agricultural Producer Barriers to Care Scale (APBCS) developed and reported by Hopkins et al., The Journal of Rural Health 41(1):e12898. This scale seems more relevant to agricultural communities and shows desirable psychometric properties. Since the psychometric properties of the BACE has not been tested in agricultural samples, was an exploratory and confirmatory factor analyses conducted to confirm the factor structure? Alpha for reliability does not measure whether the items form a construct.
A reason that agricultural operators may not seek mental health care that is rarely mentioned is that they see the challenges they face as such as the high cost of inputs, the volatility of commodity prices, weather, and all the other factors, as those that a mental health provider cannot possibly solve. They may not see how addressing their responses to these challenges is beneficial to solving the problems. It is also true that they probably do not think a behavioral health person could possibly understand what they must contend with to keep the operation viable, but there may be more to it than that. In asking about an operator’s willingness to seek or receive information on mental health, we are not asking about the reasons for their views. Previous research has shown that even when mental health access exists, farmers and ranchers to not typically seek it out.
If someone had a depression or anxiety diagnosis, that would suggest that this person sought help. This does not seem surprising because how else would they have a diagnosis if they had not talked to a medical or mental health provider?
If the BACE is designed for those with a mental health condition, you might expect it to perform differently in those with a mental health condition and those without. It might not be surprising to see significant differences in the scale and its subscales between the depression/anxiety group and those without symptoms.
The religiosity result may not be surprising given that many rural churches are doing more to address the mental health of their congregants. Some churches hire a counselor specifically to deal with mental health issues and Bible-based counseling curriculum has been developed for use in churches. Churches that cannot find the funds to have someone on staff will sometimes share a counseling across denominations. The USDA Faith-Based Initiatives have also been helping in this area. It is true that church does provide a sense of community and maybe be a buffer for those dealing with stress, anxiety, and depression, but there is more movement in directly addressing mental health by clergy in rural areas.
As the discussion mentions, more qualitative work is needed to understand the reasons why ag operators to not seek help. There is the anonymity issues and sense of independence, but there is likely to be other more complex reasons. Farmers and ranches are pragmatic people. They specialize in solving problems. They may not see how counseling is going to help them break even or make a profit in uncertain economic times. They care less about their own mental health than about the future of their farm, as the discussion correctly states.
Reviewer 2 Report
Comments and Suggestions for Authors
This manuscript is describing a cross-sectional analysis to examine Midwestern farmers’ preferences for receiving mental health information and the barriers that impede care. It is known that this population experiences higher rates of reported mental health struggles, with significant occupation-related anxiety and depression, as well as cultural norms which may diminish the legitimacy of mental health support. Although it remains unclear what organization/individual, if any, may initiate or provide mental health care access in a way that best supports Midwestern agricultural producers. This manuscript would be of interest to readers, although there are some concerns related to the statistical validity / results for some of the findings that need to be properly addressed
Abstract: no changes
Introduction: No changes
Methodology:
- Section 2.1 should be moved to the results section as you’re presenting findings prior to the reader knowing what instruments/questions were used
- Line 106: “randomly selected producers” how did you randomly determine who was selected? Every other person, every 3rd person?
Results:
- You do not directly report any sort of enrollment chart – started with 15,000 letters and ended up with ~1,000 in your sample (so about a 6.6% hit rate). How many addresses were returned as undeliverable? How many directly said no vs never responded? Did anyone withdraw their response for any reason? Any other outcomes related to enrollment?
- There is no sociodemographic table – I would take the information from methods section 2.1 (after moving to results) and make into a table that shows a visual breakdown of who was in your sample
- Lines 174-175: im not sure this analysis between sexes is statistically valid given such a dominant proportion of your sample were men (948 vs 81). If you are able to prove the statistical validity, this needs to be noted in the statistical methodology
- Lines 174 to 188: you appear to be combining total responses for very open and somewhat open in comparison to not open at all. This has some questionable statistical validity, as well as significantly diminishes the clinical relevance of your findings. It’s not surprising that a significant portion of sociodemographic variables are associated with more willingness as you’re combing two answers vs. one. I would recommend re-running this analysis and only comparing very open to not open at all for sociodemographic variables. If you are able to prove the statistical validity, this needs to be noted in the statistical methodology
- Lines 194-195: again, not sure if comparing outcomes between sexes is statistically valid given such a drastically large difference in males vs females in your sample
- Lines 194-205: this section is difficult to comprehend as well as groups together all the tables and doesn’t guide the reader throughout the layout of the findings. For example: “Mean total BACE scores as presented in Table 1. The overall barriers to care mean score was .257 (SD .42). Mean BACE scores were significantly associated with age, veteran status, and self-reported anxiety and depression symptoms and clinical diagnoses. These associations were similarly significant across the BACE subscales including the stigma subscale (Table 2), instrumental subscale (Table 3), and attitudinal subscale (Table 4). The mean for attitudinal barriers (.338, SD .49) was highest, followed by stigma-related barriers (M .248, SD .48), and instrumental barriers (M .146, SD .33).”
- Tables 1-4: I would consider re-running the analysis for openness to mental health info to separate out very vs somewhat open to mental health information. Further, these tables include a few abbreviations that are not clarified in a table footer
Discussion:
- Lines 217-219: this is an important finding, although it’s not directly stated in the results. I would add this in the paragraph of lines 159-170
- Limitations section: some major limitations were missed
- There is likely significant sampling bias as participants who experience shame, distrust of healthcare systems, or personal stigmas related to their mental health struggles may not be willing to respond to the survey OR were willing to respond and felt unable to be truly open about their perceptions of mental health
- Responses were limited to surveys structured Likert-scale type responses rather than open questions to conduct a full qualitative analysis
- It is likely that anxiety, stress, and depression symptoms and perceptions of mental health fluctuate within the year based on weather-related events, relativity to harvest/planting season, etc. so you may have experienced bias based on the time of year you sent out the surveys
Conclusion: Lines 316-317: add in “agricultural” prior to producers
